# Effective coverage of antenatal care services in post war Tigray, Northern Ethiopia: An analysis of community and health facility–based surveys

Hailay Gebretnsae[1]*, Abadi Kidanemariam Berhe[1,2], Mache Tsadik[3],
Akeza Awealom Asgedom[3], Mengistu Hagazi Tequare[1,3],
Gebregziabher Berihu Gebrekidan[3], Gebru Hailu Redae[3], Tedros Bereket[3],
Gebrekiros Gebremichael Meles[3], Mohamedawel Mohamedniguss Ebrahim[4],
Yemane Berhane Tesfau[2], Gebremedhin Gebreegziabher Gebretsadik[2],
Muzey Gebremichael Berhe[2], Hagos Degefa Hidru[1], Meresa Gebremedhin Weldu[5],
Micheale Hagos Debesay[6], Gebrehaweria Gebrekurstos[6], Rieye Esayas[6],
Haftom Gebrehiwot Woldearegay[1,7]

1 Tigray Health Research Institute, Mekelle, Tigray, Ethiopia, 2 College of Medicine and Health Sciences, Adigrat University, Adigrat, Tigray, Ethiopia, 3 School of Public Health, College of Health Sciences, Mekelle University, Mekelle, Tigray, Ethiopia, 4 School of Medicine, College of Health Sciences, Mekelle University, Mekelle, Tigray, Ethiopia, 5 School of Public Health, College of Health Science, Aksum University, Aksum, Tigray, Ethiopia, 6 Tigray Health Bureau, Mekelle, Tigray, Ethiopia, 7 College of Health Sciences, Mekelle University, Mekelle, Tigray, Ethiopia

* hailish14@gmail.com

## Abstract

### Background

Although promoting high–quality care is particularly important in post–conflict settings, little is known about the effective coverage of antenatal care (ANC) services in post war Tigray. Thus, our study was aimed to assess the effective coverage of ANC services in post war Tigray, Northern Ethiopia.

### Methods

A combined community and health facility–based cross–sectional study design was conducted in 24 randomly selected districts of Tigray, Northern Ethiopia from 29/01/2024–26/02/2024. Using multi–stage cluster sampling method, 2340 mothers of children under one year, 32 health facilities and 250 antenatal care (ANC) clients from the selected health facilities were included in the study. A pre–tested and structured questionnaire was used to collect the households' data. Additionally, checklists were used to collect data on facility readiness and process quality. Data were collected by Open Data Kit (ODK) and analyzed using SPSS version 27. The effective coverage of antenatal care (ANC) services was analyzed among the target group of women by computing the proportion who received four or more ANC visits multiplied by the average facility readiness score, received iron–folate supplementation, and the average process quality score.

**Data availability statement:** All relevant data are within the manuscript and its supporting information files.

**Funding:** The study was financially supported by Tigray Health Bureau. Its contents are solely the responsibility of the authors and do not necessarily represent the official views of the supporting offices. The funder had no role in study design, data collection and analysis, decision to publish or preparation of the manuscript.

**Competing interests:** The authors declare that they have no competing interests relevant to this manuscript.

## Results

In this study, 87.4% (95% Confidence Interval (CI): 86.1–88.8%) of women received their first antenatal care visit. However, only 10.7% (95% CI: 9.5–12.0%) of the women had their first visit before 12 weeks of gestation and the coverage of fourth and more ANC visits was 15.7% (95% CI: 14.2–17.2%). The overall ANC service readiness mean score was 55.6% (95%CI: 45.8–65.4%). Input–and intervention–adjusted ANC coverage was 8.7% and 7.1% respectively. The mean process quality score was 53.8% (95%CI: 51.0–56.6%), and the overall effective ANC coverage was 3.8%.

## Conclusions

The effective coverage of ANC services in post war Tigray is very low. To improve ANC uptake and ensure that pregnant women complete the recommended number of visits, it is crucial to enhance facility readiness by equipping essential ANC tracer items in conflict–affected Tigray region. Additionally, on-the-job training for healthcare providers working in maternal and neonatal departments is crucial to reinforce the basic components of ANC services and ensure adherence to standard protocols for delivering high quality of ANC services. Promoting early ANC initiation at health posts and encouraging pregnant women to maintain continuity in their ANC visits at nearest health centers/hospital are also vital for improving ANC4 + coverage and overall effective coverage of ANC services.

## Introduction

Antenatal care (ANC) is the care provided by health care providers to pregnant women and adolescent girls to ensure the best health conditions for both mother and baby during pregnancy [1]. Antenatal care (ANC) is a vital component of reproductive health care, aiming to reduce maternal and neonatal morbidity and mortality [1,2]. It offers an opportunity for pregnant women to engage in health promotion, screening, diagnosis, and disease prevention, empowering them to recognize potential risks, prevent complications, and prepare for childbirth [2,3]. The World Health Organization (WHO) recommends a minimum of eight ANC contacts, with the first visit taking place during the first trimester (up to 12 weeks) [2]. Attending at least four ANC visits increases the likelihood of receiving effective maternal health interventions and serves as an indicator of the quality and continuity of care during pregnancy [4].

Recently, effective ANC coverage has is a key indicator of a health system's performance, encompassing the need, utilization, and quality of services provided. It includes the coverage of contacts, facility readiness, interventions received, and the quality of services delivered [5,6]. Effective health systems have the potential to save one million newborn lives and prevent half of all maternal deaths every year [7]. In low– and middle–income countries (LMICs), poor–quality care accounts for approximately 15% of all deaths, this figure likely being even higher in conflict–affected regions [8]. However, the quality of ANC has consistently been inadequate in LMICs [9–11].

Ethiopia updated its Essential Health Service Package (EHSP) in 2019, focusing on improving service availability, accessibility, acceptability, and affordability [12]. The country is currently working to enhance quality of care at all levels, particularly to reduce preventable maternal and perinatal morbidity and mortality [13]. However, challenges such as unequal distribution of health resources, substandard quality care, and shortages of persistent essential supplies and equipment, contributing to high rates of maternal and neonatal mortality [7,12,14].

According to mini–Ethiopia Demographic and Health Survey (EDHS) 2019 report, 43% of women had at least four ANC visits [15]. However, only 12% of women received effective coverage of ANC [6]. Before the conflict, Tigray was one of the best–performing regions in maternal and child health service coverage [15,16]. Since the conflict began in November 2020, these achievements have declined drastically due to the damage and looting happened in the health facilities [17,18]. Over 70% of health facilities in Tigray were partially or completely destroyed due to the war [17]. Furthermore, the maternal mortality ratio increased from 186 per 100,000 live births before the war [19], to 840 per 100,000 live births in the region in 2021 [14].

Several previous studies in Ethiopia have focused on the general coverage of ANC services [20–25]. However, only a few studies evaluated the effective coverage of ANC services [6,26]. The findings from these studies revealed inconsistencies in the magnitude of effective coverage and variations in how was measured.

Following the signing of the Pretoria cessation of hostility agreement on November 2, 2022 [27], the Tigray Regional Health Bureau in collaboration with local and international partners, launched health recovery initiatives to rebuild the war-affected health system of Tigray. These efforts include supplying medical equipment and furniture, providing outreach health services, enhancing the capacity of healthcare providers and strengthening the overall health system. Despite these recovery efforts, the extent of effective ANC coverage in post-war Tigray remains unknown. While the term "post-war" is used to frame the study period following the formal cessation of hostilities, it is important to note that pockets of insecurity and systemic disruptions persist in some areas. Therefore, the term may not fully capture the diverse realities experienced across all zones in the region.

Promoting high-quality care is crucial, particularly in post-conflict settings [8]. Therefore, this study aims to evaluate the effective coverage of ANC services in post-conflict Tigray, Northern Ethiopia. The findings will provide crucial evidence to guide policymakers, program managers, and healthcare providers in prioritizing interventions, allocating resources, and rebuilding resilient health systems to improve ANC effective coverage in post-conflict settings.

## Methods

### Study design, setting and period

A community–and health facility–based cross–sectional study design was conducted in six accessible zones of Tigray, Northern Ethiopia, from 29/01/2024–26/02/2024. Western zone and some parts of the southern, eastern, and northwestern zones were excluded due to security concerns. The region has seven zones which are further divided into 93 districts. According to the 2007 census, the total population of Tigray in 2024 is estimated at approximately 7 million, with around 80% residing in rural areas. There region has two referral hospitals, 14 general hospitals, 24 primary hospitals, 231 health centers, and 743 health posts [28].

### Study population

The study population for the community–based study comprised all women with children under one year of age from the selected Tabias (the smallest administrative units in Ethiopia). Furthermore, the study population for the health facility–based study included ANC clients from all primary hospitals and health centers providing services to these women.

### Sample size and sampling procedure

This study was part of a large study assessment on primary healthcare services in Tigray: focused on health promotion and diseases prevention which was included maternal health services such as antenatal care, delivery, postnatal care and family planning services [29].

A total sample size of 2340 women (78 Tabias × 30 women per Tabia) who give birth within 12 months prior to data collection were included in this study. Selecting 30% of Tabias (clusters) and 20–30 HHs per Tabia (cluster) is an effective approach for achieving statistical precision while maintaining resource efficiency in household surveys using a multi-stage cluster sampling method [30–32].

A multi–stage cluster sampling method was used to select study participants for the community–based sample. In the first stage, a total of 24 districts were randomly selected from 78 accessible districts in Tigray. The second stage, 30% of the Tabias (78 Tabias in total) were randomly selected from the 24 selected districts. Finally, 30 women with children under one year of age were randomly selected from each selected Tabia (**Fig 1**). Additionally, all 32 primary health care facilities (9 primary hospitals and 23 health centers) serving the selected Tabias were included. From these facilities, a total of 250 ANC clients (5 clients from each health center and 15 clients from each primary hospital) were selected through systematic random sampling during their ANC visit (**Fig 1**).

## Data collection tools and techniques

A structured questionnaire was adapted from Ethiopian Demographic and Health Survey (EDHS) tool and other related studies [22,23,30], to collect the household (HH) data. The HH questionnaire includes socio–demographic characteristics (maternal age, marital status, maternal educational status, partner educational status, maternal occupation, partner occupation, place of residence, availability of radio and/or television in HH), reproductive health history (gravidity, parity, history of abortion, history of still birth, antenatal care (ANC), place of delivery and postnatal care (PNC)) and others related information (**S1 File**), and the data were collected through face–to–face interview.

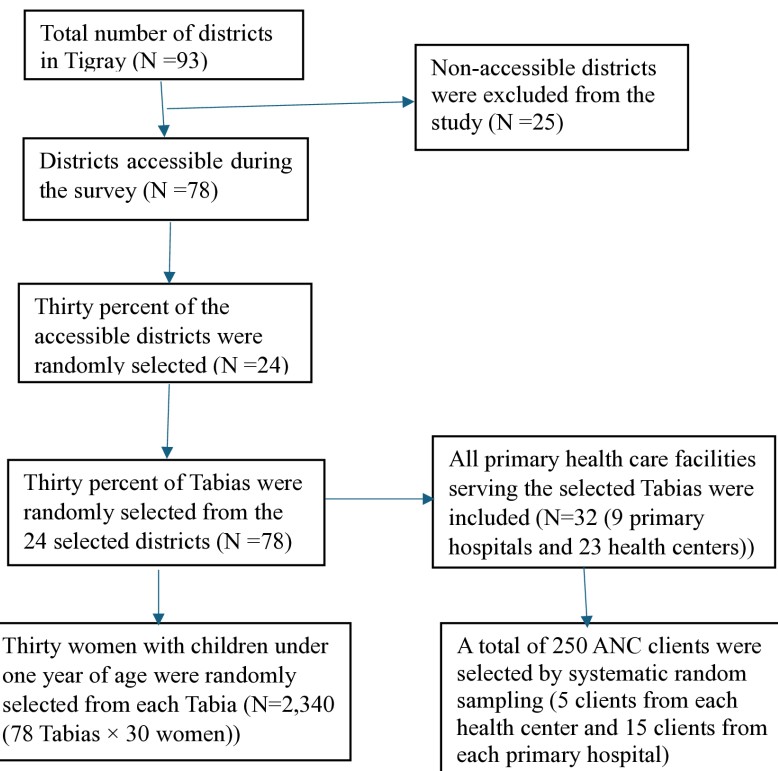

**Fig 1. Schematic presentation of the sampling procedure.**

Additionally, checklists were adapted from WHO assessment tools, national guideline and related studies [23,33–35], to collect on health facility data. Data on the availability of the ANC tracer items like: access to emergency transport, blood pressure apparatus, fetal stethoscope, adult weighing scale, examination bed, tape measure, hemoglobin tests, urine protein tests, syphilis test HIV tests kit, iron-folate tablets and tetanus toxoid vaccine were collected by interviewing health care workers at the facilities, and/or through direct observation using a predefined checklist (**S2 File**). Furthermore, ANC clients were observed during their care or procedure to check whether they received essential ANC services during their visit using observation checklist (**S3 File**). During the direct observations, confidentiality and privacy of clients were well maintained and performed without interfering with the services.

Data was collected using open data kit (ODK) template by bachelor's degree holders (nurse, midwife, or health officer) and supervised by master's degree holders in public health who had experience in collecting data for maternal health surveys.

### Data quality assurance

Data collectors and supervisors received comprehensive training for two days on the study's objectives and data collection procedures to ensure clarity and consistency of the questionnaires. Prior to commencing the actual survey, the tools were pre–tested in Adigrat town to evaluate response accuracy and estimate the required time. Based on the pre–test results, necessary adjustments like skipping pattern, coherence of the questions and options for multiple response were made correction before the actual data collection commenced. Supervisors checked the completeness and consistency of the collected data daily, and the research team maintained close communication with the data collectors and supervisors to monitor their performance. Additionally, the data collected were reviewed daily using the ODK tool from the center server.

### Measurement of variables

**Contact coverage.** This refers to the proportion of women who received four or more ANC visits.

**Facility readiness.** The readiness of health facilities to provide ANC services was assessed based on the availability of 14 tracer items: ANC guidelines, ANC checklists and/or job–aids, ANC trained staff, blood pressure apparatus, fetal stethoscope, tape measure, adult weighing scale, emergency transport, hemoglobin test, urine protein test, syphilis test, Human Immune Deficiency Virus (HIV) test kits, iron–folate tablets and tetanus toxoid vaccines. The facility readiness score was calculated using the WHO service availability and readiness assessment implementation guide [36]. Each item was assigned equal weight and recorded as a binary variable: "1" for availability and "0" for absence. The mean facility readiness score was computed by summing the available items, divided by the total number of items, and then multiplying by 100.

**Input–adjusted coverage.** This variable was computed by multiplying the proportion of women who attended four or more ANC by mean facility readiness score.

**Intervention coverage.** This refers to the proportion of women who received iron–folate supplementation.

**Intervention–adjusted coverage.** This variable was computed by multiplying the proportion who attended four or more ANC visits by mean facility readiness score and the proportion of women who received iron–folate supplementation (i.e., input–adjusted coverage × intervention coverage**).**

**Process quality score.** The process quality score was measured based on 11 essential ANC components received during ANC service (blood pressure measured, weight taken, Middle upper Arm Circumstance (MUAC) measured, fetal heartbeat listened, danger signs and symptoms of pregnancy checked and counseled, counseled on healthy diet, Hemoglobin (Hb) tested, urine analysis tested, syphilis tested, HIV tested and counseled, and preventive chemotherapy (deworming) provided). Each quality process indicator was recorded as binary variable with a value of "1 = Yes" if the client received the service and "0 = No" if the client did not receive the service. The mean process quality score was

computed by adding the received services, divided by the total number of quality process indicators, and then multiplied by 100.

**Quality–adjusted coverage.** This variable was computed by multiplying the proportion of women who attended four or more ANC visits by the mean facility readiness score, by intervention coverage, and mean process quality score (i.e., intervention–adjusted coverage × mean process quality score).

**Effective coverage.** We used quality–adjusted ANC coverage as a proxy measure of effective coverage, as data on the other two components of the effective coverage cascade (client adherence and outcome–adjusted coverage) were not available. This approach aligns with the recommendations of the "Effective Coverage Think Tank Group" [5].

## Data management and analysis

The data collected by ODK were exported to SPSS version 27 for data management and analysis. Data cleaning and checking were performed before analysis. Descriptive statistical analysis was conducted to summarize the ANC coverage by socio–demographic characteristics of study participants using frequency tables. Cross–tabulation was done to describe the frequency and percentage of services readiness and process quality components by type of health facility (health centers and primary hospitals).

To determine the contact coverage of ANC, we analyzed the proportion of women who attended four or more ANC visits. The mean facility readiness score was calculated by considering fourteen tracer items essential for providing ANC services. Input–adjusted ANC coverage was computed by multiplying the proportion of women who attended four or more ANC visits by the average facility readiness score. Intervention–adjusted coverage was determined by multiplying the proportion of women who attended four or more ANC visits, the average facility readiness score, and proportion of women who received iron–folate supplementation. Finaly, quality–adjusted ANC coverage (effective coverage of ANC) was calculated by multiplying the proportion of women who attended four or more ANC visits, the mean facility readiness score, the proportion women who received iron–folate supplementation, and the average process quality score [6].

## Ethical consideration

This study was part of a large study assessment on primary healthcare services in Tigray: focused on health promotion and diseases prevention. The study protocol was reviewed and approved by the institutional ethical review board of Tigray Health Research Institute (Reference number: THRI/4031/0503/16). Support letters were obtained from Tigray Regional Health Bureau (TRHB) and the selected district health offices. Prior to the interview, respondents were fully informed about the study's objectives and purpose, and written informed consent was obtained from each participant. Participation in the interview was voluntary, and respondents were informed of their right to withdraw at any stage. Personal identifiers were not used in the questionnaire, and confidentiality of the information was assured.

## Results

### Participants

In this study, 2338 mothers of children under one year of age participated in the household survey, achieving a 99.9% response rate. Additionally, 32 health facilities and 250 ANC clients from the selected health facilities were included.

### Coverage of ANC by socio–demographic characteristics

Out of the 2338 women surveyed, 87.4% (95% CI: 86.1–88.8%) attended their first antenatal care visit (ANC1). Hoevere, only 10.7% (95% CI: 9.5–12.0%) of the women had a first visit before 12 weeks of gestation. The coverage of fourth and more ANC visits (ANC4+) was 15.7% (95% CI: 14.2–17.2%).

A higher uptake of ANC services was observed among urban women. Urban residents reported ANC1 coverage of 92.3% and ANC4 + coverage of 29.2%, compared to rural residents, who reported ANC1 coverage of 85.5% and ANC4 + coverage of 10.6%. ANC service utilization increased significantly with maternal educational level (P–value = 0.000 for both ANC1 and ANC4+). Conversely, ANC uptake decreased as parity increased. Among women with one child (para one), the coverage of ANC1 and ANC4 + was 91.6% and 22.9%, respectively (**Table 1**).

### Reasons for not attending ANC follow–up

The main reasons for not attending ANC follow–up were lack of transport (41.0%), health facility closures (31.9%) and lack of medicines and supplies (27.1%) (**Fig 2**).

**Table 1. Distribution of ANC coverage by socio–demographic characteristics of study participants in post war Tigray, Northern Ethiopia, 2024 (n = 2338).**

| Variables | ANC1 | | | ANC4+ | | |
|---|---|---|---|---|---|---|
| | Yes (%) | No (%) | P–value) | Yes (%) | No (%) | (P–value) |
| **Place of residence** | | | | | | |
| Rural | 1452(85.5) | 246(14.5) | 0.000 | 180(10.6) | 1518(89.4) | 0.000 |
| Urban | 591(92.3) | 49(7.7) | | 187(29.2) | 453(70.8) | |
| **Age (in year)** | | | | | | |
| 18–24 | 571(90.1) | 63(9.9) | 0.002 | 104(16.4) | 530(83.6) | 0.59 |
| 25–34 | 1030(87.8) | 143(12.2) | | 187(15.9) | 986(84.1) | |
| 35–49 | 442(83.2) | 89(16.8) | | 76(14.3) | 455(85.7) | |
| **Marital status** | | | | | | |
| Currently not in union* | 258(83.8) | 50(16.2) | 0.04 | 58(18.8) | 250(81.2) | 0.11 |
| Currently in union | 1785(87.9) | 245(12.1) | | 309(15.2) | 1721(84.8) | |
| **Educational stats** | | | | | | |
| Not educated | 470(80.3) | 115(19.7) | 0.000 | 57(9.7) | 528(90.3) | 0.000 |
| Elementary | 799(88.3) | 106(11.7) | | 130(14.4) | 775(85.6) | |
| Secondary and above | 774(91.3) | 74(8.7) | | 180(21.2) | 668(78.8) | |
| **Occupation** | | | | | | |
| Housewife | 1376(88.1) | 185(11.9) | 0.098 | 242(15.5) | 1319(84.5) | 0.000 |
| Employed | 173(89.6) | 20(10.4) | | 56(29.0) | 137(71.0) | |
| Farmer | 398(84.1) | 75(15.9) | | 52(11.0) | 421(89.0) | |
| Others** | 96(86.5) | 15(13.5) | | 17(15.3) | 94(84.7) | |
| **Parity** | | | | | | |
| 1 | 460(91.6) | 42(8.4) | 0.000 | 115(22.9) | 387(77.1) | 0.000 |
| 2–4 | 1332(87.5) | 191(12.5) | | 219(14.4) | 1304(85.6) | |
| ≥5 | 251(80.2) | 62(19.8) | | 33(10.5) | 280(89.5) | |
| **Family size** | | | | | | |
| ≤5 | 930(88.8) | 117(11.2) | 0.058 | 196(18.7) | 851(81.3) | 0.000 |
| >5 | 1113(86.2) | 178(13.8) | | 171(13.2) | 1120(86.8) | |
| **Availability radio or television in the household** | | | | | | |
| Yes | 832(89.7) | 96(10.3) | 0.007 | 200(21.6) | 728(78.4) | 0.000 |
| No | 1211(85.9) | 199(14.1) | | 167(11.8) | 1243(88.2) | |

**\*Currently not in union (single, widowed, divorced and separated)**

**\*\*others (daily laborer, Merchant and student)**

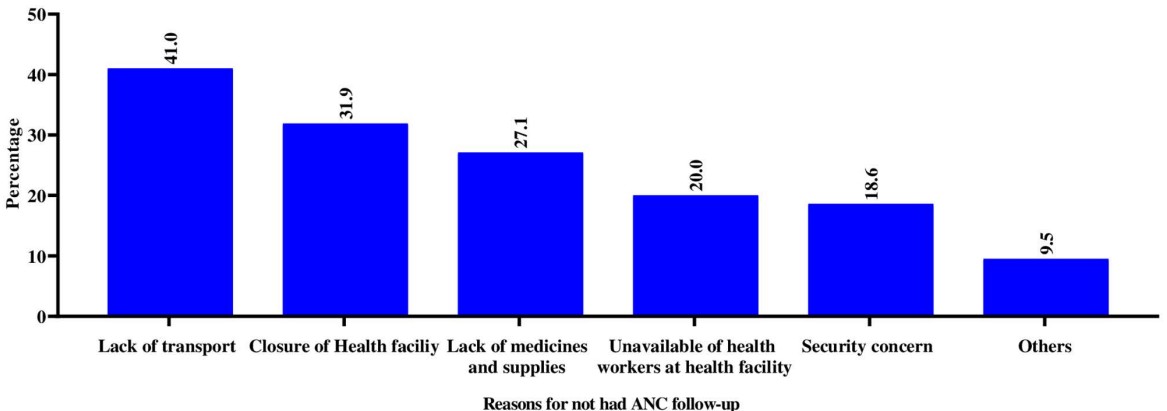

**Fig 2. Reasons for not attending ANC follow–up in post war Tigray, Northern Ethiopia, 2024 (n = 295).**

## Input–adjusted ANC coverage

Among the surveyed facilities, only 21.9% (13% of health centres and 44.4% of primary hospitals) had national ANC guidelines. Furthermore, only 31.3% of the health facilities had both ANC trained staff and access to emergency transport.

Overall, 48.1% of health centers and 74.6% of primary hospitals met the mean readiness score for ANC services (Table 2). The overall ANC service readiness mean score was 55.6% (95%CI: 45.8–65.4%). Consequently, the input–adjusted ANC coverage was calculated at 8.7% (15.7% × 55.6%).

## Intervention–adjusted ANC coverage

The coverage of women who received iron–folate supplementation was 81.4%. When multiplied by the proportion women who attended four or more ANC visits and facility readiness score, the intervention–adjusted coverage was 7.1% (81.4% × 15.7% × 55.6%).

**Table 2. Proportion of tracer items available at health facilities for providing ANC services in post war Tigray, Northern Ethiopia, 2024(n = 32).**

| Availability of ANC tracer items | Health center n (%) | Primary hospital n (%) | Total n (%) |
|---|---|---|---|
| ANC guidelines | 3(13.0) | 4(44.4) | 7(21.9) |
| ANC–checklists and/or job–aids | 8(34.8) | 5(55.6) | 13(40.6) |
| ANC trained staff | 5(21.7) | 5(55.6) | 10(31.3) |
| Blood pressure apparatus | 16(69.6) | 8(88.9) | 24(75.0) |
| Fetal stethoscope | 14(60.9) | 8(88.9) | 22(68.8) |
| Tape measure | 9(39.1) | 5(55.6) | 14(43.8) |
| Adult weighing scale | 18(78.3) | 9(100) | 27(84.4) |
| Emergency transport | 6(26.1) | 4(44.4) | 10(31.3) |
| Hemoglobin test | 12(52.2) | 7(77.8) | 19(59.4) |
| Urine protein test | 9(39.1) | 7(77.8) | 16(50) |
| Syphilis test | 8(34.8) | 7(77.8) | 15(46.9) |
| HIV test kits | 16(69.6) | 9(100) | 25(78.1) |
| Iron–folate tablets | 19(82.6) | 9(100) | 28(87.5) |
| Tetanus toxoid vaccines | 12(52.2) | 7(77.8) | 19(59.4) |
| Mean score of facility readiness (95%CI) | 48.1(36.5–59.8) | 74.6(61.1–88.1) | 55.6(45.8–65.4) |

## Quality–adjusted coverage

Among the 11 ANC quality process indicators, weight measurement, blood pressure monitoring, and fetal heartbeat checking were the most frequently reported, with the coverage ranging from 60.8–70.4%. On average, the quality process score among health centers was 41.4% (95%CI, 37.3–45.6%), while it was 64.5% (95%CI, 61.6–67.0%) among primary hospitals. The mean process quality score in this study was 53.8% (95%CI, 51.0–56.6%) (Table 3). Overall, quality–adjusted coverage was calculated by multiplying the proportion of intervention–adjusted ANC coverage by the mean quality process score. The resulting quality–adjusted coverage was 3.8% (7.1% × 53.8%).

## Effective ANC coverage

As stated in the measurement of variables, quality–adjusted ANC coverage was considered as effective coverage because information on the other two components (client adherence and outcome–adjusted coverage) were unavailable. After adjusting for facility readiness, intervention coverage, and process quality scores, the effective ANC coverage was 3.8% **(Fig 3).**

**Table 3. Proportion of process quality components received during ANC services in post war Tigray, Northern Ethiopia,2024 (n = 250).**

| Components | Health center n (%) | Primary hospital n (%) | Total n (%) |
|---|---|---|---|
| Blood pressure measured | 63(54.8) | 111(82.2) | 174(69.6) |
| Weight taken | 65(56.5) | 111(82.2) | 176(70.4) |
| MUAC measured | 49(42.6) | 34(25.2) | 83(33.2) |
| Fetal heartbeat listened | 53(46.1) | 99(73.3) | 152(60.8) |
| Danger signs and symptoms of pregnancy checked and counseled | 60(52.2) | 99(73.3) | 159(63.6) |
| Counseled on healthy diet | 70(60.9) | 79(58.5) | 149(59.6) |
| Hb tested | 32(27.8) | 72(53.3) | 104(41.6) |
| Urine analysis tested | 24(20.9) | 126(93.3) | 150(60.0) |
| Syphilis tested | 21(18.3) | 97(71.9) | 118(47.2) |
| HIV tested and counseled | 56(48.7) | 86(63.7) | 142(56.8) |
| Preventive chemotherapy (deworming) provided | 31(27.0) | 41(30.4) | 72(28.8) |
| Mean process quality score (95%CI) | 41.4(37.3–45.6) | 64.5(61.6–67.0) | 53.8(51.0–56.6) |

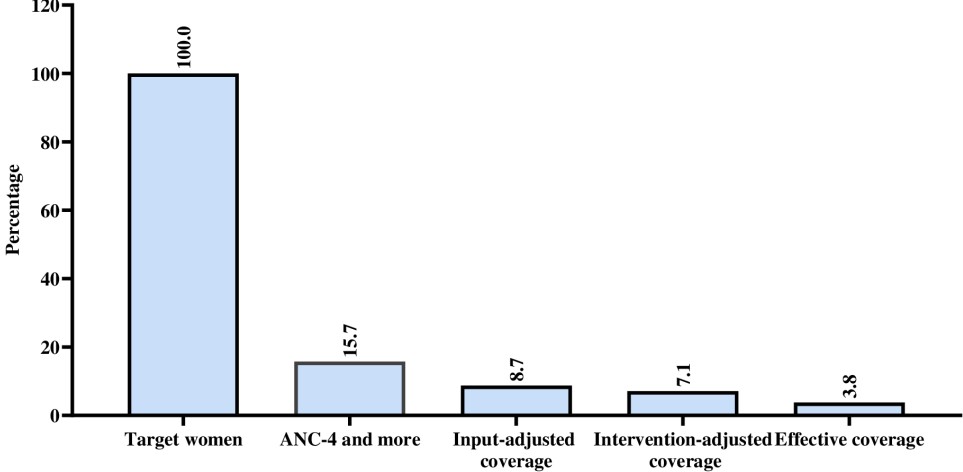

**Fig 3. Effective ANC coverage in post war Tigray, Northern Ethiopia, 2024.**

## Discussion

The findings of this study revealed that the coverage of ANC1, early ANC booking, and ANC4 + were 87.4%, 10.7%, and 15.7% respectively. The overall mean health facility readiness score was 55.6% and, 53.8% of women received the essential components of process quality based on the average score. In the effective coverage cascade, the input–adjusted, intervention–adjusted, and quality–adjusted (effective coverage) were 8.7%, 7.1%, and 3.8%, respectively.

Although the WHO recommends all women initiate ANC within 12 weeks of gestation [30], only one–tenth of women in the current study started ANC visits before 12 weeks of gestation. Our finding was lower compared to previous studies in Ethiopia, which reported early ANC booking coverages ranging from 30.5% to 57.8% [37–40], and in Sub–Saharan Africa, which reported 38.0% (ranging from 14.5% in Mozambique to 68.6% in Liberia) [41].

In this study, the coverages of ANC1 and ANC 4 + have declined compared to pre–war reports from the EDHS 2019 (ANC1: 94% and ANC4 +: 64%) [15], and Tigray statistical agency 2020 (ANC1: 90% and ANC4 +: 59%) [19]. Further-more, our finding on ANC4 + coverage was lower than those reported in other Ethiopian studies (25.3%–63.9%) [6,20–22], Tanzania (34%) [42], Zambia (60%) [43], and Pakistan (92%) [44]. This finding is supported by studies conducted in war–affected settings, which found that conflict adversely impacts maternal health service coverage [44–54].

On the other hand, ANC1 coverage (87.4%) in this study was higher compared to the period during the war (54.9%) [55]. This increase could be attributed to the implementation of health recovery initiatives following the Pretoria Cessation of Hostilities Agreement. Health facilities have resumed providing antenatal care (ANC) services, which may have encour-aged greater community health–seeking behaviour toward ANC services. However, this observation requires careful inter-pretation, as attending ANC visit does not necessarily indicate that client received all the essential components of ANC services during their visits.

Lack of transport, shortages of medicines and supplies and insufficient trained healthcare providers are barriers for the low coverage of ANC services in conflict–affected settings [56]. However, the war in Tigray was unique, marked by the killing of healthcare providers, massive displacement of people, impairment of ambulance services, disruption of health services, looting of medical supplies and equipment and widespread sexual violence [17,28,57,58]. Unlike other conflict settings, where only a few of these issues typically occur [43,50–52], these factors could be the possible reasons for the low coverage of early ANC booking and ANC4+ in Tigray.

In this study, 55.6% of health facilities achieved a mean score for ANC service availability and readiness. This find-ing was lower than pre–war figures, which were reported as 76% in Tigray and 70% nationally [6]. Primary hospitals had higher ANC readiness score compared to health centers (74.6% vs. 48.1%). This finding is consistent with previous studies, where public hospitals demonstrated higher readiness scores than health centers [6,59]. This may be attributed to partners focusing on providing equipment and medical supplies to hospitals than health centers. Another possible justifi-cation is that many of Tigray's health centers were partially or totally damaged than hospitals due to the war [17], which contributed to low quality of service.

In this study, the input–adjusted ANC coverage was 8.7%, and the intervention–adjusted ANC coverage was 7.1%. These figures were notably lower than national figures for Ethiopia, reported as 28% and 18% respectively [6]. This sug-gests women in the study area received ANC visits at health facilities without the necessary inputs and interventions. This may indicate a lack of essential services at the health facilities, potentially leading to suboptimal care for pregnant women that may end up with negative pregnancy outcomes.

Findings from this study indicate that merely attending ANC visits at health facilities does not ensure the receipt of qual-ity of care, as many women missed essential components of basic ANC services. Similar results have been reported in other studies [10]. In conflict affected areas, numerous clients expressed concerns about the poor quality of services [60]. The current study revealed a mean process quality score of 53.8%, which is lower than the 64% reported in a previous study conducted in Ethiopia [6].

Ensuring effective coverage of antenatal care (ANC) is crucial for reducing maternal and neonatal morbidity and mortality. However, this study revealed an effective ANC coverage of only 3.8%, significantly lower than pre–war Tigray (38.7%) and national levels (12%–21.9%) [6,26]. This finding was also lower than figures from other countries, such as the average effective ANC coverage across eight countries (28%) [9], Pakistan (35%) [44], India (20.9%) [61], and Rwanda (19.9%) [62].

The low effective ANC coverage observed in this study can be attributed to severe damage and inadequate facility readiness including shortages of equipment and medical supply, national guidelines, staff training, and diagnostic reagents resulting from the conflict. Studies in similar conflict–affected settings had highlighted challenges in delivering maternal and child health services due to insufficient equipment and supplies [48–50,52,54]. These issues hinder pregnant women from receiving essential ANC services during visits and disrupting the continuity of care. The quality of the first ANC visit is particularly crucial in ensuring women to complete the recommended number of ANC visits [61]. Inadequate ANC services may lead to negative experiences for pregnant women, influencing their healthcare–seeking behavior and utilization of services [35]. Low effective coverage of ANC services increases the risk of complications and adverse outcomes for pregnant women and their babies. Therefore, to ensure quality care for pregnant women through their pregnancies, it is crucial that health systems in conflict–affected areas should give priority to improving the effective ANC coverage.

## Strengths and limitations of the study

This study is the first to investigate the effective coverage of antenatal care services in the study area. The study used data from primary sources and was conducted both at the community and health facility setting which increased its reliability. However, the process quality was assessed through direct observation; therefore, healthcare providers may have improved the ANC quality of care while under observation. Additionally, we did not consider sample weighting to account for the unequal probabilities of selection associated with the multi-stage cluster sampling method. This oversight may have affected the representativeness of our findings. Furthermore, some districts were not included in this study due to security reasons, and the proportion of effective coverage of ANC in the region might be lower than the coverage estimated in this study.

## Conclusions

The effective coverage of ANC services in post war Tigray is very low, with notable deficiencies across all components of the care cascade. This study revealed a significant drop–off in continuity of care from the first to the fourth ANC visit, inadequate facility readiness and suboptimal quality of care. To improve ANC uptake and ensure that pregnant women complete the recommended number of visits, it is crucial to enhance facility readiness by equipping essential ANC tracer items in conflict-affected Tigray region. Additionally, on-the-job training for healthcare providers working in maternal and neonatal departments is crucial to reinforce the basic components of ANC services and ensure adherence to standard protocols for delivering high-quality ANC care. Promoting early ANC initiation at health posts and encouraging pregnant women to maintain continuity in their ANC visits at nearest health center/hospital are also vital for improving ANC4 + coverage and overall effective coverage of ANC services. Moreover, more comprehensive research is needed to fully address the effective coverage of the entire continuum of maternal health care services.

## Supporting information

**S1 File.  This S1 File questionnaire which was used to collect a community –based data from women.**
(PDF)

**S2 File.  This S2 File checklist which was used to collect a health facility–based data.**
(PDF)

**S3 File. This S3 File checklist which was used to collect a process quality data from ANC clients.**
(PDF)

**S4 File. This S1 Checklist contains the STROBE check list of the manuscript.**
(DOCX)

## Acknowledgments

We express our sincere gratitude to Tigray Health Bureau and the selected District Health Offices for providing technical support. We also thank for the supervisors, data collectors and study participants for their contributions to the successful completion of this research.

## Author contributions

**Conceptualization:** Hailay Gebretnsae, Abadi Kidanemariam Berhe, Mache Tsadik, Haftom Gebrehiwot Woldearegay.

**Data curation:** Hailay Gebretnsae, Abadi Kidanemariam Berhe, Gebrekiros Gebremichael Meles, Mohamedawel Mohamedniguss Ebrahim, Haftom Gebrehiwot Woldearegay.

**Formal analysis:** Hailay Gebretnsae.

**Funding acquisition:** Hailay Gebretnsae, Mache Tsadik.

**Investigation:** Hailay Gebretnsae.

**Methodology:** Hailay Gebretnsae, Abadi Kidanemariam Berhe, Mache Tsadik, Akeza Awealom Asgedom, Mengistu Hagazi Tequare, Gebregziabher Berihu Gebrekidan, Gebrekiros Gebremichael Meles, Mohamedawel Mohamedniguss Ebrahim, Yemane Berhane Tesfau, Gebremedhin Gebreegziabher Gebretsadik, Muzey Gebremichael Berhe, Hagos Degefa Hidru, Meresa Gebremedhin Weldu, Haftom Gebrehiwot Woldearegay.

**Project administration:** Mache Tsadik, Micheale Hagos Debesay, Gebrehaweria Gebrekurstos, Rieye Esayas.

**Resources:** Mache Tsadik, Micheale Hagos Debesay, Gebrehaweria Gebrekurstos, Rieye Esayas.

**Software:** Hailay Gebretnsae, Gebrekiros Gebremichael Meles, Mohamedawel Mohamedniguss Ebrahim, Muzey Gebremichael Berhe.

**Supervision:** Hailay Gebretnsae, Abadi Kidanemariam Berhe, Mache Tsadik, Akeza Awealom Asgedom, Mengistu Hagazi Tequare, Gebregziabher Berihu Gebrekidan, Gebru Hailu Redae, Tedros Bereket, Yemane Berhane Tesfau, Gebremedhin Gebreegziabher Gebretsadik, Muzey Gebremichael Berhe, Hagos Degefa Hidru, Meresa Gebremedhin Weldu, Haftom Gebrehiwot Woldearegay.

**Validation:** Hailay Gebretnsae, Abadi Kidanemariam Berhe, Mache Tsadik, Akeza Awealom Asgedom, Mengistu Hagazi Tequare, Gebregziabher Berihu Gebrekidan, Gebru Hailu Redae, Tedros Bereket, Mohamedawel Mohamedniguss Ebrahim, Yemane Berhane Tesfau, Gebremedhin Gebreegziabher Gebretsadik, Hagos Degefa Hidru, Meresa Gebremedhin Weldu, Haftom Gebrehiwot Woldearegay.

**Visualization:** Hailay Gebretnsae, Abadi Kidanemariam Berhe, Haftom Gebrehiwot Woldearegay.

**Writing – original draft:** Hailay Gebretnsae.

**Writing – review & editing:** Hailay Gebretnsae, Abadi Kidanemariam Berhe, Mache Tsadik, Akeza Awealom Asgedom, Mengistu Hagazi Tequare, Gebregziabher Berihu Gebrekidan, Gebru Hailu Redae, Tedros Bereket, Gebrekiros Gebremichael Meles, Mohamedawel Mohamedniguss Ebrahim, Yemane Berhane Tesfau, Gebremedhin Gebreegziabher Gebretsadik, Muzey Gebremichael Berhe, Hagos Degefa Hidru, Meresa Gebremedhin Weldu, Micheale Hagos Debesay, Gebrehaweria Gebrekurstos, Rieye Esayas, Haftom Gebrehiwot Woldearegay.

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
