## [Decision Letter · Decision Letter 0]

18 Nov 2024

PONE-D-24-39443Effective coverage of antenatal care services in post war Tigray, Northern Ethiopia: An analysis of community and health facility-based surveysPLOS ONE

Dear Dr. Gebretnsae,

Thank you for submitting your manuscript to PLOS ONE. After careful consideration, we feel that it has merit but does not fully meet PLOS ONE’s publication criteria as it currently stands. Therefore, we invite you to submit a revised version of the manuscript that addresses the points raised during the review process.

We look forward to receiving your revised manuscript.

Kind regards,

Awol Yemane Legesse

Academic Editor

PLOS ONE

Journal Requirements: When submitting your revision, we need you to address these additional requirements. 1. Please ensure that your manuscript meets PLOS ONE's style requirements, including those for file naming. The PLOS ONE style templates can be found at https://journals.plos.org/plosone/s/file?id=wjVg/PLOSOne_formatting_sample_main_body.pdf and https://journals.plos.org/plosone/s/file?id=ba62/PLOSOne_formatting_sample_title_authors_affiliations.pdf 2. We note that the grant information you provided in the ‘Funding Information’ and ‘Financial Disclosure’ sections do not match.  When you resubmit, please ensure that you provide the correct grant numbers for the awards you received for your study in the ‘Funding Information’ section.

**Additional Editor Comments:**

I congratulate the research team for conducting such a big study.

The paper requires drastic improvement for English language. There are multiple issues such as typos, grammatic errors, word usage, and formatting. I recommend you seek proof reading from proficient individual.

Background

The entire background needs to be rewritten, the AI score is 90%.

Please improve the flow of the background,

Lines 68-73 have complementary ideas but you have tried to connect them as if they were conflicting.

Please address this in your introductions. How can health care providers and policy makers effectively mitigate the challenges in resource and quality of care in the region. What strategies have been implemented to improve effective ANC coverage in war affected areas like Tigray.

Methods

You are advised to cite the source for the information regarding population data. For instance, Line 108-111.

Please cite the study, or if you are unable to do so, remove the statements 118-120.

The sample size and sampling procedure is difficult to decipher for the readers. Schematic presentation, i.e figure, would make it easier. Furthermore, you have to explain why you selected 8 urban/16 rural. Why 30% Tabias, and why 30 women.

‘. In the first stage, a total of 24 districts (8 urban and 16 rural) were randomly selected from 78 accessible districts of Tigray. The second stage, 30% Tabias were selected randomly (a total of 78 tabias) from the 24 selected districts. Finally, 30 women with under one year child were randomly selected from each selected Tabia’

Why 32 primary hospitals and what is the basis for the proportion of 9 primary hospitals vs 23 health center. How did you come up with the 250 ANC clients, and the proportion of 5 to 15 ? Justify it/ cite reference.

ANC 4, It would help if you mentioned it as a limitation, why most of your metrics fall within the old ANC recommendations of 4 visits instead of contacts of the revised WHO recommendations for positive pregnancy experience.

Discussion

The discussion needs a massive upgrade. It is important to showcase the importance of the study more precisely. Currently, the argument utilized in the discussion is shallow/superficial. A more detailed comparison with pre war conditions is important. Furthermore, please compare it with other conflict ridden areas. Most importantly, emphasize on showcasing how the impact of the Tigray war varies from the effects of other wars on ANC utilization and ANC quality. Please factor in the dynamics of the war in this context.

Conclusions

You have to stick to your study findings. Some of the recommendations put forward are from preexisting knowledge that predates your findings.

What is the implication of your study?

Please attend to the comments by the reviewers.

Reviewers' comments:

Reviewer's Responses to Questions

**Comments to the Author**

1. Is the manuscript technically sound, and do the data support the conclusions?

Reviewer #1: Partly

Reviewer #2: Yes

2. Has the statistical analysis been performed appropriately and rigorously? 

Reviewer #1: No

Reviewer #2: Yes

3. Have the authors made all data underlying the findings in their manuscript fully available?

Reviewer #1: Yes

Reviewer #2: Yes

4. Is the manuscript presented in an intelligible fashion and written in standard English?

Reviewer #1: No

Reviewer #2: Yes

5. Review Comments to the Author

Reviewer #1: First, I would like to congratulate the authors for conducting this important study, which aims to determine the state of antenatal care (ANC) in the post-war period.

However, there are some grammatical issues that should be addressed throughout the manuscript.

Methods

This study uses multistage survey data. The authors employed a multistage cluster sampling technique to select districts, tabiyas, and households. Typically, multistage surveys require sample weighting to account for unequal probabilities of selection. Based on the selection criteria described, not all households had an equal chance of being included in the study. Applying sample weighting would adjust for this unequal probability and improve the representativeness of the findings. If feasible, I recommend incorporating sample weighting to make the results more reflective of the target population.

Results

Page 12, line 242: The subtitle reads, “Reasons for not had ANC follow-up.” Please revise this to correct the grammar. Additionally, ensure consistency in language usage throughout the manuscript. For example, while both "centre" and "center" are correct spellings, they belong to British and American English, respectively. Consistency in style is recommended.

Discussion

The study was conducted in early 2024, at least a year after the war ended. An ANC coverage of 87.4% is not particularly low and warrants further explanation in the discussion. For example, what was the ANC1 coverage during the war (very likely it was much lower than 87.4%)? What actions were taken in the one-year post-war period that led to this coverage to 87.4% level? Although it is lower than the pre-war period, it is still relatively high, and understanding these dynamics would enhance the discussion.

Strengths and Limitations

The main limitation of this study is that it did not adjust for potential confounders, which may have contributed to the observed lower ANC coverage during the post-war period. This limitation should be clearly mentioned.

Conclusions

Page 19, line 352: The conclusion is not fully supported by the findings. The authors state that inadequate facility readiness and suboptimal quality of care contributed to the lower ANC coverage, but no analysis was conducted to support this claim. I suggest revising the conclusion to align with the results.

Reviewer #2: Effective coverage of antenatal care services in post war Tigray, Northern Ethiopia: An

analysis of community and health facility-based surveys

Overall comment

• I am delighted to come across with the manuscript. It was organized very well

Comments

• As the WHO currently revise the schedule for 8 ANC visit, but the author uses 4 visits? What is your justification.

• As this study was done in a post recovery from the war, dealing on quality could be matter. However, it was cognizant that 70% of the health facilities were damaged or looted?

Abstract

• On the background section, the authors described that … Although promoting high-quality care is particularly important in post-conflict settings, little is known. However, in post conflict setting despite the quality of the service matter but the priority should be to enhance access for the service as the service was dramatically damaged due to the war.

• In line with this, what is the real gap for conducting such study should be described in a brief manner.

• On the method section, there are two populations

o 2340 mothers of children under one year

o 250 antenatal care (ANC) clients from the selected health facilities were included in

What is the difference between these two populations?

o Add the type of analysis employed to answer the objective of the study

o The research put an output for the Input-and intervention-adjusted ANC and what about for the contact coverage and Quality adjusted coverage related to the variable?

Results

• Do you have quality-adjusted antenatal coverage estimates?

Introduction

• Overall, it was organized very well. However, what could be the justification for emphasising on quality than in access in area where the 70% of the health facilities were not functional or partially functional.

Methods

The authors described that this study was part of a large study. Does the large study published? If yes add the reference rather than describing a sampling procedure of the Ethiopia demographic health survey

• Do you have a sampling frame to select the eligible women as what they did in the EDHS?

• Does the tool part of the large study or you develop a tool specifically for this study?

• On the measurement section to describe the variable of interest under your investigation?

o Intervention coverage. This refers to the proportion of women who received iron-folate tablets.? what about TT vaccination

Results

• Can you organize the table 2 and 3 information using graph?

Discussion

• How conducting a study in community and health facility setting which increased its reliability.

• What mechanism does the authors uses to reduce Howerton effect bias?

• What type of observation, does the authors use to observe for the process quality?

• The exclusion of some districts as result of security issue mentioned as limitation of the study. Is that real limitation?

o Do you have methodological limitation like not including TT vaccination in the measurement of intervention coverage.

On conclusion

• The first paragraph…. You conclude that he overall effective coverage of ANC services in post war Tigray is very low

o It was obvious as the study was done in post conflict setting.

Introduction

Methods

Results

Discussion

Conclusion and recommendations

6. PLOS authors have the option to publish the peer review history of their article (what does this mean? ). If published, this will include your full peer review and any attached files.

**Do you want your identity to be public for this peer review?** For information about this choice, including consent withdrawal, please see our Privacy Policy .

Reviewer #1: No

Reviewer #2: **Yes: ** Mussie Alemayehu

---

## [Author Response · Author response to Decision Letter 1]

29 Dec 2024

Responses to editor’s and reviewers’ comments

Editor ’s comment

Journal Requirements:

Editor ’s comment

Response to editor’s comment: Thank you for your guidance, we have tried to revise our manuscript based on the PLOSE ONE journal style template and journal requirements.

Editor ’s comment

Response to editor’s comment: Thank you for your insightful comment. However, the study was supported by per diem and vehicle support provided by Tigray Health Bureau. There is no formal grant number associated with this support.

Additional Editor Comments:

Editor ’s comment: The paper requires drastic improvement for English language. There are multiple issues such as typos, grammatic errors, word usage, and formatting. I recommend you seek proof reading from proficient individual.

Response to editor’s comment: We are grateful for your comment. We have made significant language corrections as advised.

Background

Editor ’s comment: The entire background needs to be rewritten, the AI score is 90%.

Please improve the flow of the background,

Response to editor’s comment: Thank you, we have rewritten the background as advised.

Editor ’s comment: Lines 68-73 have complementary ideas but you have tried to connect them as if they were conflicting.

Response to editor’s comment: Thank you, we have revised the paragraph accordingly.

Editor ’s comment: Please address this in your introductions. How can health care providers and policy makers effectively mitigate the challenges in resource and quality of care in the region. What strategies have been implemented to improve effective ANC coverage in war affected areas like Tigray.

Response to editor’s comment: Thank you for your insightful comment. We have added as suggested.

The revised introduction section on page 5, paragraph 6 and line 99-103 now reads, “Following the signing of the Pretoria cessation of hostility agreement on November 2, 2022[27], the Tigray Regional Health Bureau in collaboration with local and international partners, launched health recovery initiatives to rebuild the war-affected health system of Tigray. These efforts include supplying medical equipment and furniture, providing outreach health services, enhancing the capacity of healthcare providers and strengthening the overall health system.”

Methods

Editor ’s comment: You are advised to cite the source for the information regarding population data. For instance, Line 108-111.

Response to editor’s comment: Thank you for your comment. I have added a reference as suggested.

Editor ’s comment: Please cite the study, or if you are unable to do so, remove the statements 118-120.

Response to editor’s comment: Thank you for your comment. We added a reference as suggested.

Editor ’s comment: The sample size and sampling procedure is difficult to decipher for the readers. Schematic presentation, i.e figure, would make it easier. Furthermore, you have to explain why you selected 8 urban/16 rural. Why 30% Tabias, and why 30 women. In the first stage, a total of 24 districts (8 urban and 16 rural) were randomly selected from 78 accessible districts of Tigray. The second stage, 30% Tabias were selected randomly (a total of 78 tabias) from the 24 selected districts. Finally, 30 women with under one year child were randomly selected from each selected Tabia’

Response to editor’s comment: Thank you, we have added Schematic presentation by figure in the revised manuscript as suggested.

Regarding the selection districts, Tabias and eligible women. We did not intentionally select 8 urban and 16 rural. We randomly selected 24 districts from the 76 accessible districts, which resulted in a selection of 8 urban and 16 rural districts by chance. Regarding our selection of 30% of Tabias and 30 women with children under one year old, we recognize that including the entire population or a large sample size is ideal for representativeness. However, due to resource constraints, this approach is not feasible. Therefore, selecting 30% of the sample size is a feasible that balances representativeness with practicality in our research design. Additionally, we treated each Tabia as a cluster and included 30 households per Tabia/cluster due to these limited resources. As we have detailed in the sample size and sampling procedure section, selecting 20-30 households per enumeration area/cluster is considered adequate for our study [27].

Editor ’s comment: Why 32 primary hospitals and what is the basis for the proportion of 9 primary hospitals vs 23 health center. How did you come up with the 250 ANC clients, and the proportion of 5 to 15 ? Justify it/ cite reference.

Response to editor’s comment: Our objective is to link the community-based study with a health facility-based study to evaluate the effective coverage of antenatal care (ANC) services. To achieve this, we included all 32 primary health care facilities (9 primary hospitals and 23 health centers) that serve the selected Tabias to assess health facility readiness. Additionally, we purposively sampled 250 ANC clients based on expert consensus and then we selected an average of 5 clients from each health center and 15 clients from each primary hospital by systematic random sampling by considering the ANC client load from the previous three months to assess ANC quality process components.

Editor ’s comment: ANC 4, It would help if you mentioned it as a limitation, why most of your metrics fall within the old ANC recommendations of 4 visits instead of contacts of the revised WHO recommendations for positive pregnancy experience.

Response to editor’s comment: Thank you for your comment. However, the revised WHO recommendations, which suggest at least 8 contacts during pregnancy have not yet been fully implemented in the Tigray region due to the conflict. Furthermore, effective coverage of ANC services can be calculated without requiring ANC4 or ANC8; it can also be determined using the coverage of ANC1 (26).

Discussion

Editor ’s comment: The discussion needs a massive upgrade. It is important to showcase the importance of the study more precisely. Currently, the argument utilized in the discussion is shallow/superficial. A more detailed comparison with pre war conditions is important. Furthermore, please compare it with other conflict ridden areas. Most importantly, emphasize on showcasing how the impact of the Tigray war varies from the effects of other wars on ANC utilization and ANC quality. Please factor in the dynamics of the war in this context.

Response to editor’s comment: Thank you, we have done major modification in discussion section as suggested in the revised manuscript.

Conclusions

Editor ’s comment: You have to stick to your study findings. Some of the recommendations put forward are from preexisting knowledge that predates your findings.

What is the implication of your study?

Response to editor’s comment: Thank you for your comment. We have aligned our recommendations based on our findings as suggested.

Reviewers' comments:

Reviewer #1

Methods

Reviewer’s comment: This study uses multistage survey data. The authors employed a multistage cluster sampling technique to select districts, tabiyas, and households. Typically, multistage surveys require sample weighting to account for unequal probabilities of selection. Based on the selection criteria described, not all households had an equal chance of being included in the study. Applying sample weighting would adjust for this unequal probability and improve the representativeness of the findings. If feasible, I recommend incorporating sample weighting to make the results more reflective of the target population.

Response to review’s comment: Thank for your observation. Yes, of course, it recommended to use sapling weight in multistage cluster sampling design, However, it not always strict requirements because in our case we assume that the study population had qual probabilities of being selected and the same characteristics. In our sampling strategy, all individuals or households within the sampled Tabias were selected with equal probability using simple random technique. As such, there was no unequal selection probability to adjust for, eliminating the need for sample weights.

Results

Reviewer’s comment: Page 12, line 242: The subtitle reads, “Reasons for not had ANC follow-up.” Please revise this to correct the grammar. Additionally, ensure consistency in language usage throughout the manuscript. For example, while both "centre" and "center" are correct spellings, they belong to British and American English, respectively. Consistency in style is recommended.

Response to reviewer’s comment: Thank you, the comment is well accepted. We have edited the “Reasons for not had ANC follow-up” to “Reasons for not attending ANC follow-up” and we changed word “centre" to “center" in revised manuscript.

Discussion

Reviewer’s comment: The study was conducted in early 2024, at least a year after the war ended. An ANC coverage of 87.4% is not particularly low and warrants further explanation in the discussion. For example, what was the ANC1 coverage during the war (very likely it was much lower than 87.4%)? What actions were taken in the one-year post-war period that led to this coverage to 87.4% level? Although it is lower than the pre-war period, it is still relatively high, and understanding these dynamics would enhance the discussion.

Response to reviewer’s comment: Thank you for your insightful comment. We have incorporated the comment in the revised manuscript as suggested. The revised discussion section on page 17, paragraph 4 and line 307-313 now reads,” On the other hand, ANC1 coverage (87.4%) in this study was higher compared to the period during the war (54.9%) [53]. This increase could be attributed to the implementation of health recovery initiatives following the Pretoria Cessation of Hostilities Agreement. Health facilities have resumed providing antenatal care (ANC) services, which may have encouraged greater community health–seeking behaviour toward ANC services. However, this observation requires careful interpretation, as attending ANC visit does not necessarily indicate that client received all the essential components of ANC services during their visits”.

Strengths and Limitations

Reviewer’s comment: The main limitation of this study is that it did not adjust for potential confounders, which may have contributed to the observed lower ANC coverage during the post-war period. This limitation should be clearly mentioned.

Response to reviewer’s comment: Thank you for your comment. However, our primary objective in this study is to assess the effective coverage of antenatal care (ANC) services. Identifying factors that contribute to lower ANC coverage is not our objective. Effective coverage is a relatively new concept defined as the proportion of a population in need of a service that gains a positive health outcome from that service (5). To calculate the effective coverage of ANC, we simply multiplied the proportion of women who received four or more ANC visits by the mean facility readiness score, the proportion of women who received iron-folate, and the average process quality score (6).

Conclusions

Reviewer’s comment: Page 19, line 352: The conclusion is not fully supported by the findings. The authors state that inadequate facility readiness and suboptimal quality of care contributed to the lower ANC coverage, but no analysis was conducted to support this claim. I suggest revising the conclusion to align with the results.

Response to reviewer’s comment: Thank you for your comment. It is important to clarify that our conclusion pertains to the low effective coverage of antenatal care (ANC) services, which is different from the low ANC coverage. Normally, we could not identify the factors affecting for low effective coverage of ANC services by statistical analysis using bivariate and multi-variable model analysis. However, as we have clearly stated in method section effective coverage of ANC services was calculated by multiplying the proportion of women who received four or more ANC visits (15.7%) by the mean of health facility readiness score (55.6%), the proportion women who received iron-folate (81.4%), and the average process quality score (53.8%).

I.e. effective coverage of ANC services=15.7%*55.6%*81.4%*53.8%= (0.157*0.556*0.814*0.538) * 100% =3.8%.

The above calculation indicates that effective coverage of ANC services is affected by coverage of ANC4+, mean of health facility readiness score, coverage of iron-folate supplementation, and ANC basic services process quality score.

• ANC1 coverage =87.4% and ANC4+=15.7%. Therefore, there is a significant drop-off in continuity of care from the first to the fourth ANC visit (from 87.4% to15.7%.).

• Mean of health facility readiness score=55.6% it indicates inadequate facility readiness.

• Mean of basic ANC quality process score=53.8% it indicates suboptimal quality of ANC service.

Therefore, with the above justification, we believe that our conclusion is based on our findings. “The overall effective coverage of ANC services in post war Tigray is very low, with deficiencies observed across each component of the cascade. The primary factors contributing to this low effective coverage of ANC services include a significant drop-off in continuity of care from the first to the fourth ANC visit, as well as inadequate facility readiness and suboptimal quality of care service”.

Reviewer #2

Reviewer’s comment: • As the WHO currently revise the schedule for 8 ANC visit, but the author uses 4 visits? What is your justification.

Response to reviewer’s comment: Thank you for your comment. However, the revised WHO recommendations, which suggest at least 8 contacts during pregnancy have not yet been fully implemented in the Tigray region due to the conflict. Furthermore, effective coverage of ANC services can be calculated without requiring ANC4 or ANC8; it can also be determined using the coverage of ANC1 (26).

Reviewer’s comment: • As this study was done in a post recovery from the war, dealing on quality could be matter. However, it was cognizant that 70% of the health facilities were damaged or looted?

Response to reviewer’s comment: Thank you for your comment. As we mentioned in the methods section, this study is part of the large study which was included the services utilization and effective coverage because promoting high-quality care is particularly important in post-conflict settings (8).

Abstract

Reviewer’s comment: • On the background section, the authors described that … Although promoting high-quality care is particularly important in post-conflict settings, little is known. However, in post conflict setting despite the quality of the service matter but the priority should be to enhance access for the service as the service was dramatically damaged due to the war.

• In line with this, what is the real gap for conducting such study should be described in a brief manner.

Response to reviewer’s comment: Thank you for your insightful comment. Absolutely true, access to services is a top priority in post-war settings. However, in our assessment of effective coverage, we also evaluated service access and contact coverage. Access to health services/ health facility readiness is key components of effective coverage.

Reviewer’s comment: • On the method section, there are two populations

o 2340 mothers of children under one year

o 250 antenatal care (ANC) clients from the selected health facilities were included in

What is the difference between these two populations?

Response to reviewer’s comment: Our objective is to link the community-based study with a health facility-based study to evaluate the effective coverage of antenatal care (ANC) services. To achieve this, we included 2340 mothers of children under one year to assess crude ANC coverage (coverage of ANC+), we included 32 primary health care facilities to assess the health facility readiness, and we include

---

## [Decision Letter · Decision Letter 1]

2 Feb 2025

PONE-D-24-39443R1Effective coverage of antenatal care services in post war Tigray, Northern Ethiopia: An analysis of community and health facility–based surveysPLOS ONE

Dear Dr. Gebretnsae,

Thank you for submitting your manuscript to PLOS ONE. After careful consideration, we feel that it has merit but does not fully meet PLOS ONE’s publication criteria as it currently stands. Therefore, we invite you to submit a revised version of the manuscript that addresses the points raised during the review process.

We look forward to receiving your revised manuscript.

Kind regards,

Kindu Yinges Wondie, MSc

Academic Editor

PLOS ONE

Journal Requirements:

Additional Editor Comments:

Comments on Manuscript:

"Effective coverage of antenatal care services in post war Tigray, Northern Ethiopia: An analysis of community and health facility–based surveys"

Thank you, the authors, for the great job. Here are my concerns I suggest you to revise:

1. Introduction

# The introduction provides adequate background on the importance of ANC and the challenges in post-conflict settings. However, the emphasis on quality over access in the introduction may seem contradictory given the context where 70% of health facilities were damaged or looted. Clarify how the study aims to balance these priorities.

# Ensure the research gap is clearly articulated. For example, why is effective coverage of ANC particularly relevant in this context compared to contact coverage or access alone?

2. Methods

# The methods section is detailed but could benefit from more organization. For instance, separating sampling procedures for the community based and health facility based components into distinct subsections would improve readability.

# The tools adapted from EDHS and WHO should be briefly described.

# Some details (e.g., exclusion of districts due to security issues) are mentioned but not fully explored in terms of their potential impact on the findings.

2.1 Sampling Technique

# The use of a multi stage cluster sampling method is appropriate for a study of this scope and geography. However, your justification for selecting 30% of Tabias and 30 households per Tabia could be made more clear. While resource constraints are valid, citing additional evidence or methodological guidelines (beyond referencing the EDHS standards) would strengthen this decision.

# The claim that 8 urban and 16 rural districts were selected "by chance" could use more elaboration, as this may raise concerns about the balance and representation of the sampling frame.

# please ensure clarity on the randomization process used at each stage (districts, Tabias, households). A flowchart was added as per the editor’s suggestion, which helps, but detailed descriptions would enhance transparency.

2.2 Sample Size Determination

# While the manuscript provides some rationale for the sample size (e.g., EDHS standards and resource constraints), a clearer power calculation or justification based on the study’s primary outcomes would strengthen the methodological rigor.

# The selection of 250 ANC clients (15 per primary hospital and 5 per health center) appears arbitrary. The justification provided (based on client load) is insufficient and should be supported with references or additional details.

# Please genuinly address why weighting was not applied to account for multi stage sampling, as noted by the reviewers. Even if you assume equal probabilities of selection, explicitly discussing this decision would preempt concerns about representativeness.

3.Discussion

# The discussion is repetitive in places and could be restructured for clarity. For instance, comparisons with pre-war data and other conflict-affected regions should be consolidated into one cohesive argument rather than scattered across multiple paragraphs.

# While the authors highlight the uniqueness of the Tigray conflict, they could delve deeper into how this impacts ANC differently compared to other conflict settings. For example, the widespread looting and destruction of health facilities could be linked to specific quality gaps observed in the study.

4. Recommendations

# The recommendations lack specificity in some areas. For instance, while promoting early ANC booking is essential, practical strategies to address barriers (e.g., transport issues, health facility readiness) could be more explicitly stated.

# Suggestions for training healthcare providers are too broad. Details about the specific gaps in provider competencies and the type of training required would make it actionable.

# The call to improve facility readiness is valid, but given resource constraints in post conflict settings, prioritization of specific interventions could be emphasized.

# Recommendations should clearly align with the findings of the study. For example, the observed drastic drop off from ANC1 to ANC4+ coverage should be linked to targeted interventions to address this continuity gap.

This study provides valuable insights into effective ANC coverage in a challenging post-war setting. By addressing the above points, particularly around sampling, recommendations, and methodological clarity, the manuscript can significantly enhance its rigor and impact.

Reviewers' comments:

Reviewer's Responses to Questions

**Comments to the Author**

1. If the authors have adequately addressed your comments raised in a previous round of review and you feel that this manuscript is now acceptable for publication, you may indicate that here to bypass the “Comments to the Author” section, enter your conflict of interest statement in the “Confidential to Editor” section, and submit your "Accept" recommendation.

Reviewer #1: (No Response)

Reviewer #3: (No Response)

2. Is the manuscript technically sound, and do the data support the conclusions?

Reviewer #1: Yes

Reviewer #3: Yes

3. Has the statistical analysis been performed appropriately and rigorously? 

Reviewer #1: Yes

Reviewer #3: Yes

4. Have the authors made all data underlying the findings in their manuscript fully available?

Reviewer #1: Yes

Reviewer #3: Yes

5. Is the manuscript presented in an intelligible fashion and written in standard English?

Reviewer #1: Yes

Reviewer #3: Yes

6. Review Comments to the Author

Reviewer #1: Dear Reviewers,

Thank you for addressing all my comments, except one, which I still don't think is fully addressed.

The authors responded the following "In our sampling strategy, all individuals or households within the sampled Tabias were selected with equal probability using simple random technique."

However, inside the document, I did not see a section that states simple random sampling was used. Also, the only way all the participants could have equal probability of being selected without using sample weighting is, if you conduct simple random sampling of the individuals, which is impossible, because it is not feasible to list all individuals from 24 districts. All districts have different number of tabyas and all tabyas have different number of household, and all households have different number of individuals, which is why sample weighting is very crucial here. So, I suggest, if sample weighting is not feasible at this time, the authors should at least mention this as a limitation.

Reviewer #3: Dear editor,

I would like to thank the editor for the opportunity to review this interesting manuscript. After carefully evaluating the study, I find that it has merit; however, I recommend the following revisions to enhance its clarity and rigor.

Comments and Suggestions for the Authors:

1. Sampling Technique: The methodology regarding participant selection is not clearly described. The authors should provide a detailed explanation of the sampling process and illustrate the number of participants recruited from each district using a diagram.

2. Data Quality Assurance: The manuscript states that the questionnaire underwent a pre-test in a different location to assess response accuracy and estimate the required time. Please specify the location where the pre-test was conducted and describe any adjustments made based on the pre-test results.

3. The results are well-articulated and clearly presented.

4. Given the reported low effective coverage of antenatal care (ANC) services in post-war Tigray, it is important to specify the baseline used for comparison. Clarifying this will help contextualize the findings.

5. Strengths and Limitations: The section discussing the strengths and limitations of the study is not well-developed. A thorough revision is needed to clearly highlight the key strengths and potential limitations.

6. It would be beneficial for the authors to suggest directions for future research based on the study’s findings.

7. The conclusion does not fully reflect the study’s key findings. A thorough revision is needed to ensure alignment with the results and key discussions.

8. References: DOIs should be included for all referenced articles where applicable

7. PLOS authors have the option to publish the peer review history of their article (what does this mean? ). If published, this will include your full peer review and any attached files.

**Do you want your identity to be public for this peer review?** For information about this choice, including consent withdrawal, please see our Privacy Policy .

Reviewer #1: No

Reviewer #3: No

---

## [Author Response · Author response to Decision Letter 2]

14 Mar 2025

Editor’s comments:

1. Introduction

Editor ’s comment: The introduction provides adequate background on the importance of ANC and the challenges in post-conflict settings. However, the emphasis on quality over access in the introduction may seem contradictory given the context where 70% of health facilities were damaged or looted. Clarify how the study aims to balance these priorities.

Response to editor’s comment: Thank you for your comment. You are right, ensuring access to ANC services is a top priority in post-war environments, where 70% of health facilities have been damaged or looted. However, it is equally important to promote high-quality care in these post-conflict settings (8). In our assessment, we incorporated various dimensions to evaluate effective coverage, including access to health services/health facility readiness, contact coverage, intervention coverage, and quality processes. Since the readiness of health facilities is a key component of effective coverage, we believe our study adequately addresses the access to ANC services as well.

Editor ’s comment: Ensure the research gap is clearly articulated. For example, why is effective coverage of ANC particularly relevant in this context compared to contact coverage or access alone?

Response to editor’s comment: Thank you for your insightful comment. However, contact coverage or access is nothing without high quality of care. Additionally, the components of effective coverage are facility readiness, contact coverage, intervention coverage and process quality. Therefore, we have assessed effective coverage and all its components as the time.

2. Methods

Editor ’s comment: The methods section is detailed but could benefit from more organization. For instance, separating sampling procedures for the community based and health facility-based components into distinct subsections would improve readability.

Response to editor’s comment: Thank you for your valuable comment. In our study, the community-based and health facility-based components are linked through the sampling procedures, as the chosen health facilities are located within the selected Tabias that serve the catchment population.

Editor ’s comment: The tools adapted from EDHS and WHO should be briefly described.

Response to editor’s comment: Thank you for your insightful comment. We have described as suggested.

The revised methods section (data collection tools and techniques sub-section) on page7, and line 145-158 now reads, “The HH questionnaire includes socio–demographic characteristics (maternal age, marital status, maternal educational status, partner educational status, maternal occupation, partner occupation, place of residence, availability of radio and/or television in HH), reproductive health history (gravidity, parity, history of abortion, history of still birth, antenatal care (ANC), place of delivery and postnatal care (PNC)) and others related information (S1 File), and the data were collected through face–to–face interview.

Additionally, checklists were adapted from WHO assessment tools, national guideline and related studies [23,31–33], to collect on health facility data. Data on the availability of the ANC tracer items like: access to emergency transport, blood pressure apparatus, fetal stethoscope, adult weighing scale, examination bed, tape measure, hemoglobin tests, urine protein tests, syphilis test HIV tests kit, iron-folate tablets and tetanus toxoid vaccine were collected by interviewing health care workers at the facilities, and/or through direct observation using a predefined checklist (S2 File)”.

Editor ’s comment: Some details (e.g., exclusion of districts due to security issues) are mentioned but not fully explored in terms of their potential impact on the findings.

Response to editor’s comment: Thank you for your comment, we have already mentioned the potential impact on the limitation section on page 20, and line 375-378 now reads, Furthermore, some districts were not included in this study due to security reasons, and the proportion of effective coverage of ANC in the region might be lower than the coverage estimated in this study.

2.1 Sampling Technique

Editor ’s comment: The use of a multistage cluster sampling method is appropriate for a study of this scope and geography. However, your justification for selecting 30% of Tabias and 30 households per Tabia could be made more clear. While resource constraints are valid, citing additional evidence or methodological guidelines (beyond referencing the EDHS standards) would strengthen this decision.

Response to editor’s comment: Thank you for your valuable comment. We have added additional references as suggested.

The revised methods section (Sample size and sampling procedure sub-section) on page 6, and line 128-132 now reads, “Selecting 30% of Tabias (clusters) and 20–30 HHs per Tabia (cluster) is an effective approach for achieving statistical precision while maintaining resource efficiency in household surveys using a multi-stage cluster sampling method [30–32]”.

Editor ’s comment: The claim that 8 urban and 16 rural districts were selected "by chance" could use more elaboration, as this may raise concerns about the balance and representation of the sampling frame.

Response to editor’s comment: Thank you for your comment. We have lists of all 78 accessible districts in Tigray and then we randomly selected 24 districts using the lottery method to include in the study and finally 8 urban and 16 rural districts were selected.

Editor ’s comment: please ensure clarity on the randomization process used at each stage (districts, Tabias, households). A flowchart was added as per the editor’s suggestion, which helps, but detailed descriptions would enhance transparency.

Response to editor’s comment: Thank you for your comment, but we have already included as fig 1 in the revised manuscript.

2.2 Sample Size Determination

Editor ’s comment: While the manuscript provides some rationale for the sample size (e.g., EDHS standards and resource constraints), a clearer power calculation or justification based on the study’s primary outcomes would strengthen the methodological rigor.

Response to editor’s comment: Thank you, we have added some justification in the revised manuscript.

Editor ’s comment: The selection of 250 ANC clients (15 per primary hospital and 5 per health center) appears arbitrary. The justification provided (based on client load) is insufficient and should be supported with references or additional details.

Response to editor’s comment: Thank you for your comment. Honestly, we did not use any references in the selection of 250 ANC clients (15 per primary hospital and 5 per health center). We planned to include a sample of 250 ANC clients based on an expert consensus because of cost effectiveness and then we selected an average of 5 clients from each health center and 15 clients from each primary hospital by systematic random sampling by considering the ANC client load from the previous three months to assess ANC quality process components.

Editor ’s comment: Please genuinly address why weighting was not applied to account for multi stage sampling, as noted by the reviewers. Even if you assume equal probabilities of selection, explicitly discussing this decision would preempt concerns about representativeness.

Response to editor’s comment: Thank you for your valuable feedback. We have addressed the issue of sample weighting in the revised limitations section, as suggested by the reviewers.

The revised methods discussion (strengths and limitations of the study sub-section) on page 20, and nd line 373-375 now reads, “Additionally, we did not consider sample weighting to account for the unequal probabilities of selection associated with the multi-stage cluster sampling method. This oversight may have affected the representativeness of our findings”.

3.Discussion

Editor ’s comment: The discussion is repetitive in places and could be restructured for clarity. For instance, comparisons with pre-war data and other conflict-affected regions should be consolidated into one cohesive argument rather than scattered across multiple paragraphs.

Response to editor’s comment: Thank you for your comment. As you may have noticed, we present multiple indicators related to antenatal care (ANC) service utilization, including ANC1, early initiation of ANC, and ANC4, among others. Consequently, we have tried to discuss each indicator in the context of pre-war data and findings from other conflict-affected areas, which is why you may see multiple information presented in various sections.

Editor ’s comment: While the authors highlight the uniqueness of the Tigray conflict, they could delve deeper into how this impacts ANC differently compared to other conflict settings. For example, the widespread looting and destruction of health facilities could be linked to specific quality gaps observed in the study.

Response to editor’s comment: Thank you for your comment, but we have already included in the consecutive sentence. These factors could be the possible reasons for the low coverage early ANC booking and ANC4+ in the region.

4. Recommendations

Editor ’s comment: The recommendations lack specificity in some areas. For instance, while promoting early ANC booking is essential, practical strategies to address barriers (e.g., transport issues, health facility readiness) could be more explicitly stated.

Response to editor’s comment: Thank you for your comment. We have tried to modify our recommendations as you suggested.

Editor ’s comment: Suggestions for training healthcare providers are too broad. Details about the specific gaps in provider competencies and the type of training required would make it actionable.

Response to editor’s comment: Thank you for your insightful comment. We have revised as you suggested.

Editor ’s comment: The call to improve facility readiness is valid, but given resource constraints in post conflict settings, prioritization of specific interventions could be emphasized.

Response to editor’s comment: Thank you for your insightful comment. We have revised as suggested.

Editor ’s comment: Recommendations should clearly align with the findings of the study. For example, the observed drastic drop off from ANC1 to ANC4+ coverage should be linked to targeted interventions to address this continuity gap.

Response to editor’s comment: Thank you for your comment. We have tried to link our findings with our recommendation as suggested.

Reviewers' comments:

Reviewer #1

Reviewer’s comment: The authors responded the following "In our sampling strategy, all individuals or households within the sampled Tabias were selected with equal probability using simple random technique." However, inside the document, I did not see a section that states simple random sampling was used. Also, the only way all the participants could have equal probability of being selected without using sample weighting is, if you conduct simple random sampling of the individuals, which is impossible, because it is not feasible to list all individuals from 24 districts. All districts have different number of tabyas and all tabyas have different number of household, and all households have different number of individuals, which is why sample weighting is very crucial here. So, I suggest, if sample weighting is not feasible at this time, the authors should at least mention this as a limitation.

Response to reviewer’s comment: Thank you for your insightful comment. We have incorporated the issue of sample weighting into the revised limitations section, as you suggested.

Reviewer #3

Reviewer’s comment: 1. Sampling Technique: The methodology regarding participant selection is not clearly described. The authors should provide a detailed explanation of the sampling process and illustrate the number of participants recruited from each district using a diagram.

Response to reviewer’s comment: Thank you for your comment, but we have already included the sampling procedure by diagram in Fig 1.

Reviewer’s comment: 2. Data Quality Assurance: The manuscript states that the questionnaire underwent a pre-test in a different location to assess response accuracy and estimate the required time. Please specify the location where the pre-test was conducted and describe any adjustments made based on the pre-test results.

Response to reviewer’s comment: Thank you, the comment is well accepted. We have mentioned the location which pre-test was conducted and the adjustments we made based on the pre-test results.

The revised methods discussion (data quality assurance sub-section) on page 8, and line 168-171 now reads, “Prior to commencing the actual survey, the tools were pre–tested in Adigrat town to evaluate response accuracy and estimate the required time. Based on the pre–test results, necessary adjustments like skipping pattern, coherence of the questions and options for multiple response were made correction before the actual data collection commenced”.

Reviewer’s comment: 3. The results are well-articulated and clearly presented.

Response to reviewer’s comment: Thank you very much for you recognition to our results.

Reviewer’s comment: 4. Given the reported low effective coverage of antenatal care (ANC) services in post-war Tigray, it is important to specify the baseline used for comparison. Clarifying this will help contextualize the findings.

Response to reviewer’s comment: Thank you for your comment, but we have already compared with pre–war in Tigray (38.7%) and national levels (12%–21.9%) [6,26].

Reviewer’s comment: 5. Strengths and Limitations: The section discussing the strengths and limitations of the study is not well developed. A thorough revision is needed to clearly highlight the key strengths and potential limitations.

Response to reviewer’s comment: Thank you for your comment. We have made a revision of the strengths and limitations section accordingly.

Reviewer’s comment:6. It would be beneficial for the authors to suggest directions for future research based on the study’s findings.

Response to reviewer’s comment: Thank you for your valuable comment. We have recommended future research in the revised manuscript.

Reviewer’s comment: 7. The conclusion does not fully reflect the study’s key findings. A thorough revision is needed to ensure alignment with the results and key discussions.

Response to reviewer’s comment: Thank you for your comment. We have revised as you suggested.

Reviewer’s comment: 8. References: DOIs should be included for all referenced articles where applicable

Response to reviewer’s comment: Thank you for your valuable comment. We have included DOIs as you suggested.

---

## [Decision Letter · Decision Letter 2]

24 Jul 2025

PONE-D-24-39443R2Effective coverage of antenatal care services in post war Tigray, Northern Ethiopia: An analysis of community and health facility–based surveysPLOS ONE

Dear Dr. Gebretnsae,

Thank you for submitting your manuscript to PLOS ONE. After careful consideration, we feel that it has merit but does not fully meet PLOS ONE’s publication criteria as it currently stands. Therefore, we invite you to submit a revised version of the manuscript that addresses the points raised during the review process.

We look forward to receiving your revised manuscript.

Kind regards,

Kindu Yinges Wondie, MSc

Academic Editor

PLOS ONE

Journal Requirements:

**Additional Editor Comments:**

Academic Editor Decision on PONE-D-24-39443R2

First and fore most I want to appreciate the authors for dealing this important component of the maternal continuum of care which often gets disruptions in areas affected by war.

I accept the publication of your manuscript with minor revision for the following concerns.

# the first concern is the use of the term "post-war Tigray". While it is well justified in the contextual timeline of the study, caution must be taken given that the conflict dynamics were continuing throughout the period of this study. This means while open hostilities may have officially ended, insecurity persists in some zones. Thus, Using “post-war” may oversimplify ongoing humanitarian and systemic challenges including incomplete health system recovery. If the authors want to retain the term “post-war”, I suggest adding a clarification sentence in the Introduction or Methods section. Or they may acknowledge that the post-war label may not universally apply across all districts.

# the second concern is in the measurement of effective coverage. While the omission of outcome-adjusted and adherence-adjusted coverage in the measurement of the effective coverage is clearly acknowledged, they are advised to acknowledge sensitivity of the multiplicative method (ANC4+ × readiness × intervention coverage × process quality) to small changes in any variable, which may lead to very low estimates (3.8% in this case). In addition giving each component the equal weight may not reflect their real-world impact (e.g., process quality may matter more than one missing tracer item). The use of Iron-folate supplementation as the sole intervention in the calculation of "intervention coverage" also need clarification as it may underestimate the broader scope of ANC content. In addition giving each component the equal weight may not reflect their real-world impact (e.g., process quality may matter more than one missing tracer item). The use of Iron–folate supplementation as the sole intervention in the calculation of "intervention coverage" also may underestimate the broader scope of ANC content.

Reviewers' comments:

Reviewer's Responses to Questions

**Comments to the Author**

1. If the authors have adequately addressed your comments raised in a previous round of review and you feel that this manuscript is now acceptable for publication, you may indicate that here to bypass the “Comments to the Author” section, enter your conflict of interest statement in the “Confidential to Editor” section, and submit your "Accept" recommendation.

Reviewer #1: All comments have been addressed

Reviewer #3: All comments have been addressed

Reviewer #4: (No Response)

2. Is the manuscript technically sound, and do the data support the conclusions?

Reviewer #1: Yes

Reviewer #3: Yes

Reviewer #4: Yes

3. Has the statistical analysis been performed appropriately and rigorously? 

Reviewer #1: Yes

Reviewer #3: Yes

Reviewer #4: Yes

4. Have the authors made all data underlying the findings in their manuscript fully available?

Reviewer #1: Yes

Reviewer #3: Yes

Reviewer #4: Yes

5. Is the manuscript presented in an intelligible fashion and written in standard English?

Reviewer #1: Yes

Reviewer #3: Yes

Reviewer #4: Yes

6. Review Comments to the Author

Reviewer #1: The authors has addressed all my comments. I believe that the manuscript is now suitable for publication.Thank you

Reviewer #3: thank you the author.The comments I provided earlier have been addressed, and I have no further comments regarding the effective coverage of antenatal care among pregnant women in post-war Tigray.

Reviewer #4: The issue it investigates sound great.

Comments:

1. A justification statement that appear at line 98-100, which indicates the inconsistency of the previous study findings, is not an appropriate justification for a primary study. Such kinds of justifications will be addressed by conducting a review (like Systematic review and/or Meta analysis) of the previous inconsistent studies not by primary study like the current one.

2. The introduction section better address the significance of the study.

3. You used very old data, almost 20 years ago, to indicate the population size of Tigray (line 116). I would rather recommend to use the estimated one (if available) since we do not have a recent census.

4. You better use an appropriate English term for "Tabias" at line 121 and other lines. It can be "Sites" or any other.

5. The study population, specially facility based study population, are vague as well as contradictory with the consequent descriptions. It is stated as if the facilities are the study population (line 121-123) but it is not like that as stated in the next sections and according to the general essence of the study.

6. Your sampling procedure is unclear.

Eg. line 128: "..... 2340 women (78 Tabias x 30 Households per tabia).......". The computation in the bracket will give number of households in the the 78 Tabias not the number of women. Or did you assume 1 women in a household?

7. You better specify the type of observation you used at line 160-161. Is it overt or covert(concealed)? If it is overt, how did you manage the bias?

7. PLOS authors have the option to publish the peer review history of their article (what does this mean? ). If published, this will include your full peer review and any attached files.

**Do you want your identity to be public for this peer review?** For information about this choice, including consent withdrawal, please see our Privacy Policy .

Reviewer #1: No

Reviewer #3: No

Reviewer #4: **Yes: ** Mengistu Abate Belay

---

## [Author Response · Author response to Decision Letter 3]

11 Aug 2025

Responses to editor’s and reviewers’ comments

We are writing to express our sincere gratitude to both the editor and the reviewers for your invaluable and insightful comments. Your feedback has been instrumental in significantly enhancing the quality of our manuscript to reach this stage.

We have carefully addressed all comments and incorporated the necessary revisions. Our manuscript has also been formatted according to the PLOS ONE style guidelines to ensure full compliance. Accordingly, we are pleased to resubmit a revised version of the manuscript and revised manuscript with tracked changes for your easy reference, along with this a detailed point-by-point response letter addressing each comment.

Thank you very much, looking forward to your favorable response

Below is our point-by-point response to the editor’s and reviewers’ comments for your kind consideration.

Editor comments:

Editor ’s comment: 1. # the first concern is the use of the term "post-war Tigray". While it is well justified in the contextual timeline of the study, caution must be taken given that the conflict dynamics were continuing throughout the period of this study. This means while open hostilities may have officially ended, insecurity persists in some zones. Thus, Using “post-war” may oversimplify ongoing humanitarian and systemic challenges including incomplete health system recovery. If the authors want to retain the term “post-war”, I suggest adding a clarification sentence in the Introduction or Methods section. Or they may acknowledge that the post-war label may not universally apply across all districts.

Response to editor’s comment:

Thank you for this important and thoughtful observation. We acknowledge that referring to the period as “post-war” may risk oversimplifying the complex and evolving security and humanitarian situation in Tigray. However, we have used the term “post-war” specifically in reference to the period following the signing of the Pretoria Peace Agreement in November 2022, which formally marked the cessation of hostilities.

To address this concern, we have added a clarifying sentence in the Introduction page 5 and lines number 107 -110), now reads, “…. While the term “post-war” is used to frame the study period following the formal cessation of hostilities, it is important to note that pockets of insecurity and systemic disruptions persist in some areas. Therefore, the term may not fully capture the diverse realities experienced across all zones in the region.”

Editor ’s comment: 2. # the second concern is in the measurement of effective coverage. While the omission of outcome-adjusted and adherence-adjusted coverage in the measurement of the effective coverage is clearly acknowledged, they are advised to acknowledge sensitivity of the multiplicative method (ANC4+ × readiness × intervention coverage × process quality) to small changes in any variable, which may lead to very low estimates (3.8% in this case). In addition giving each component the equal weight may not reflect their real-world impact (e.g., process quality may matter more than one missing tracer item). The use of Iron-folate supplementation as the sole intervention in the calculation of "intervention coverage" also need clarification as it may underestimate the broader scope of ANC content. In addition giving each component the equal weight may not reflect their real-world impact (e.g., process quality may matter more than one missing tracer item). The use of Iron–folate supplementation as the sole intervention in the calculation of "intervention coverage" also may underestimate the broader scope of ANC content

Response to editor’s comment:

We sincerely appreciate this valuable feedback. We agree that the multiplicative method for estimating effective coverage is inherently sensitive to small variations in any of its components (ANC4+, facility readiness, intervention coverage, and process quality), which can result in a lower overall estimate. This aligns with the intuitive calculations of effective coverage as recommended by the Measurement of Effective Coverage Think Tank Group (1). Furthermore, World Health Organization (WHO) Service Availability and Readiness Assessment (SARA) uses equal weighting when it calculates facility reediness and quality process (2). In fact, studies have shown that equal weighting and expert-based weighting approaches yield similar index scores; however, because expert ratings can vary across expert groups, the equal-weighting method is generally preferred due to its simplicity, replicability, and endorsement by WHO (3).

While we acknowledge that assigning equal weights to all components may not fully reflect their relative real-world impact, we adopted this approach in the absence of standardized or validated weighting schemes in the literature. Equal weighting also facilitates comparability with other studies and ensures a pragmatic and transparent methodology.

Regarding the use of iron–folate supplementation for intervention coverage, we selected it as the sole proxy indicator for ANC intervention coverage based on three considerations: (1) the consistent availability of iron–folate supplementation data across all gestational ages in the dataset; (2) its universal recommendation by the World Health Organization (WHO) as a critical component of ANC; and (3) its widespread implementation and documentation, which facilitates comparability across settings.

We recognize, however, that relying solely on iron–folate supplementation may underestimate the full spectrum of ANC content, as it excludes other essential interventions. To our knowledge, there is no strict requirement to use multiple interventions for calculating effective coverage; alternatively, it can be estimated without selecting a specific intervention by multiplying the average quality of service by crude (contact) coverage (4).

To support our approach, we refer to the following key sources:

1. Marsh AD, Muzigaba M, Diaz T, Requejo J, Jackson D, Chou D, et al. (2020). Effective coverage measurement in maternal, newborn, child, and adolescent health and nutrition: progress, future prospects, and implications for quality health systems. Lancet Global Health, 8(5).

2. WHO. Service Availability and Readiness Assessment (SARA): An annual monitoring system for service delivery implementation Guide. World Heal Organ ;2015.

3. World Health Organization. (2015). Tracking universal health coverage: First global monitoring report. Geneva: World Health Organization and the World Bank.

4. Yakob B, Gage A, Nigatu TG, Hurlburt S, Hagos S, Dinsa G, et al. Low effective coverage of family planning and antenatal care services in Ethiopia. Int J Qual Heal Care. 2019;31(10):725–32.

Reviewers' comments:

Reviewer #4:

Reviewer’s comment: 1. A justification statement that appear at line 98-100, which indicates the inconsistency of the previous study findings, is not an appropriate justification for a primary study. Such kinds of justifications will be addressed by conducting a review (like Systematic review and/or Meta analysis) of the previous inconsistent studies not by primary study like the current one.

Response to reviewer’s comment:

Thank you for your valuable comment. We agree that when inconsistencies in previous study findings are observed, a systematic review or meta-analysis is generally the more appropriate approach to address such discrepancies. In our manuscript, we referred to the inconsistency as part of describing the overall knowledge gap, noting that the available evidence is both scarce and varied. However, our primary aim was not to assess the consistency of previous findings. Instead, the main objective of our study was to evaluate the effective coverage of ANC services in the post-war context of the Tigray Region, where no sufficient primary data currently exists.

Reviewer’s comment: 2. The introduction section better address the significance of the study.

Response to reviewer’s comment: Thank you for your insightful comment. We have added as suggested.

The revised introduction section on page 5 and line 112-115 now reads, “The findings will provide crucial evidence to guide policymakers, program managers, and healthcare providers in prioritizing interventions, allocating resources, and rebuilding resilient health systems to improve ANC effective coverage in post-conflict settings.”

Reviewer’s comment: 3. You used very old data, almost 20 years ago, to indicate the population size of Tigray (line 116). I would rather recommend to use the estimated one (if available) since we do not have a recent census.

Response to reviewer’s comment: Thank you for your comment. We have rephrased the sentence accordingly. We have used the projections from the 2007 census because there is not census after 2007 in Ethiopia.

The revised methods section and Study design, setting and period sub section on page 6 and line 121-123 now reads, “According to the 2007 census, the total population of Tigray in 2024 is estimated at approximately 7 million, with around 80% residing in rural areas.”

Reviewer’s comment: 4. You better use an appropriate English term for "Tabias" at line 121 and other lines. It can be "Sites" or any other.

Response to reviewer’s comment: Thank you for your comment. We have revised as you suggested “Tabias (the smallest administrative units in Ethiopia).”

Reviewer’s comment: 5. The study population, specially facility based study population, are vague as well as contradictory with the consequent descriptions. It is stated as if the facilities are the study population (line 121-123) but it is not like that as stated in the next sections and according to the general essence of the study.

Response to reviewer’s comment: Thank you observation. We have revised as the following in the revised manuscript “the study population for the health facility–based study included ANC clients from all primary hospitals and health centers providing services to these women.”

Reviewer’s comment: 6. Your sampling procedure is unclear.

Eg. line 128: "..... 2340 women (78 Tabias x 30 Households per tabia).......". The computation in the bracket will give number of households in the the 78 Tabias not the number of women. Or did you assume 1 women in a household?

Response to reviewer’s comment: Thank you for your insightful comment. We have changed the households to women in the revised methods section and Sample size and sampling procedure sub section on page 6 and line 131-133 now reads, “A total sample size of 2340 women (78 Tabias × 30 women per Tabia) who give birth within 12 months prior to data collection were included in this study.”

Reviewer’s comment: 7. You better specify the type of observation you used at line 160-161. Is it overt or covert(concealed)? If it is overt, how did you manage the bias?

Response to reviewer’s comment:

Thank you for your insightful comment. The observation conducted in our study was overt, meaning that participants were aware that they were being observed. To minimize observation bias, we ensured that data collectors were trained to be as unobtrusive and neutral as possible during the observation process. Additionally, the observers maintained consistent behavior across all sessions to reduce the biases. We have already stated this in the in methods section and data collection tools and techniques sub section in the revised manuscript.

---

## [Decision Letter · Decision Letter 3]

21 Oct 2025

Effective coverage of antenatal care services in post war Tigray, Northern Ethiopia: An analysis of community and health facility–based surveys

PONE-D-24-39443R3

Dear Dr. Gebretnsae,

We’re pleased to inform you that your manuscript has been judged scientifically suitable for publication and will be formally accepted for publication once it meets all outstanding technical requirements.

Kind regards,

Miquel Vall-llosera Camps

Staff Editor

PLOS One

Reviewers' comments:

Reviewer's Responses to Questions

**Comments to the Author**

1. If the authors have adequately addressed your comments raised in a previous round of review and you feel that this manuscript is now acceptable for publication, you may indicate that here to bypass the “Comments to the Author” section, enter your conflict of interest statement in the “Confidential to Editor” section, and submit your "Accept" recommendation.

Reviewer #1: All comments have been addressed

Reviewer #4: All comments have been addressed

2. Is the manuscript technically sound, and do the data support the conclusions?

Reviewer #1: Yes

Reviewer #4: Yes

3. Has the statistical analysis been performed appropriately and rigorously? 

Reviewer #1: Yes

Reviewer #4: Yes

4. Have the authors made all data underlying the findings in their manuscript fully available?

Reviewer #1: Yes

Reviewer #4: Yes

5. Is the manuscript presented in an intelligible fashion and written in standard English?

Reviewer #1: Yes

Reviewer #4: Yes

6. Review Comments to the Author

Reviewer #1: (No Response)

Reviewer #4: I do not have any comments for now. The authors adress all my concerns.

7. PLOS authors have the option to publish the peer review history of their article (what does this mean? ). If published, this will include your full peer review and any attached files.

**Do you want your identity to be public for this peer review?** For information about this choice, including consent withdrawal, please see our Privacy Policy .

Reviewer #1: No

Reviewer #4: **Yes: ** Mengistu Abate Belay

---

## [Editor Report · Acceptance letter]

PONE-D-24-39443R3

PLOS ONE

Dear Dr. Gebretnsae,

I'm pleased to inform you that your manuscript has been deemed suitable for publication in PLOS ONE. Congratulations! Your manuscript is now being handed over to our production team.

Kind regards,

on behalf of

Dr. Miquel Vall-llosera Camps

Staff Editor

PLOS ONE